# Avoiding the Pitfalls of siRNA Delivery to the Retinal Pigment Epithelium with Physiologically Relevant Cell Models

**DOI:** 10.3390/pharmaceutics12070667

**Published:** 2020-07-16

**Authors:** Eva Ramsay, Manuela Raviña, Sanjay Sarkhel, Sarah Hehir, Neil R. Cameron, Tanja Ilmarinen, Heli Skottman, Jørgen Kjems, Arto Urtti, Marika Ruponen, Astrid Subrizi

**Affiliations:** 1Drug Research Program, Division of Pharmaceutical Biosciences, Faculty of Pharmacy, University of Helsinki, 00014 Helsinki, Finland; eva.ramsay@helsinki.fi (E.R.); manuela.ravina@gmail.com (M.R.); sanjay.sarkhel@helsinki.fi (S.S.); arto.urtti@uef.fi (A.U.); 2Department of Life Sciences, Institute of Technology Sligo, F91 YW50 Sligo, Ireland; hehir.sarah@itsligo.ie; 3School of Engineering, University of Warwick, Coventry CV4 7AL, UK; neil.cameron@monash.edu; 4Department of Materials Science and Engineering, Monash University, Clayton, VIC 3800, Australia; 5Faculty of Medicine and Health Technology, BioMediTech, Tampere University, 33014 Tampere, Finland; tanja.ilmarinen@tuni.fi (T.I.); heli.skottman@tuni.fi (H.S.); 6Interdisciplinary Nanoscience Center (iNANO), Aarhus University, 8000 Aarhus C, Denmark; jk@mbg.au.dk; 7Laboratory of Biohybrid Technologies, Institute of Chemistry, St. Petersburg State University, 198504 Peterhoff, Russia; 8School of Pharmacy, University of Eastern Finland, 70210 Kuopio, Finland; marika.ruponen@uef.fi

**Keywords:** siRNA delivery, polyplex, lipoplex, lipidoid, retinal pigment epithelium, physiologically relevant RPE cell model, melanin, melanosome

## Abstract

Inflammation is involved in the pathogenesis of several age-related ocular diseases, such as macular degeneration (AMD), diabetic retinopathy, and glaucoma. The delivery of anti-inflammatory siRNA to the retinal pigment epithelium (RPE) may become a promising therapeutic option for the treatment of inflammation, if the efficient delivery of siRNA to target cells is accomplished. Unfortunately, so far, the siRNA delivery system selection performed in dividing RPE cells in vitro has been a poor predictor of the in vivo efficacy. Our study evaluates the silencing efficiency of polyplexes, lipoplexes, and lipidoid-siRNA complexes in dividing RPE cells as well as in physiologically relevant RPE cell models. We find that RPE cell differentiation alters their endocytic activity and causes a decrease in the uptake of siRNA complexes. In addition, we determine that melanosomal sequestration is another significant and previously unexplored barrier to gene silencing in pigmented cells. In summary, this study highlights the importance of choosing a physiologically relevant RPE cell model for the selection of siRNA delivery systems. Such cell models are expected to enable the identification of carriers with a high probability of success in vivo, and thus propel the development of siRNA therapeutics for ocular disease.

## 1. Introduction

Age-related macular degeneration (AMD), diabetic retinopathy, and glaucoma are common age-related eye conditions and major causes of visual impairment and blindness. According to the World Health Organization, the number of people affected by AMD and glaucoma will increase from 195.6 million in 2020 to 243.4 million in 2030 due to a variety of factors such as ageing population, lifestyle, and noncommunicable diseases (WHO world report on vision). Although the underlying mechanisms responsible for the pathogenesis of AMD have not been fully elucidated, the presence of inflammatory cells in the ocular tissues of AMD patients had been recognized already in the 1980s [1,2]. Considerable evidence now supports the central role of chronic low-level inflammation (para-inflammation) in the pathogenesis of several human diseases [3], including age-related eye disorders such as AMD [4,5,6,7], diabetic retinopathy [8,9,10,11], and glaucoma [12].

AMD is triggered by the degeneration of the retinal pigment epithelium (RPE), a highly specialized monolayer of pigmented, terminally differentiated, post-mitotic cells joined apically with tight junctions that form the outer blood-retina barrier (BRB) [13]. This barrier regulates the traffic of cells and molecules between the blood and the neural retina, and it thus plays a crucial role in maintaining the viability of the retina. The RPE exerts several essential supportive functions of homeostasis, including the daily phagocytosis of photoreceptor outer segments, light absorption to optimize vision and protect from photo-oxidative damage, the upkeep of the visual cycle, and the secretion of growth factors essential for the maintenance of the retina and the choriocapillaris [14]. The RPE also plays an essential role in modulating the immune-regulatory environment of the eye by protecting it from an exaggerated immune response that may cause harmful chronic inflammation and a variety of pathologic processes. RPE dysfunction can negatively affect the retina in two ways; first, it may lead to the breakdown of the BRB and to the consequent loss of the immune privilege of the eye. Second, the RPE can also contribute to the establishment of a proinflammatory environment in the neural retina by secreting a variety of inflammatory cytokines, chemokines, and growth factors [15,16]. A therapeutic strategy that silences the proinflammatory genes in the RPE may thus alleviate the inflammation and restore retinal homeostasis.

Gene silencing can be achieved with small interfering RNA (siRNA). siRNA is a powerful tool, whose selective and sequence-specific gene knockdown has been demonstrated in a variety of tissues in vitro and in vivo [17,18] and, in 2018, it has led to the approval of the first siRNA, patisiran, for the treatment of polyneuropathy in patients with hereditary transthyretin amyloidosis [19]. The eye is an ideal target organ for siRNA therapeutics because, given its immune privileged and enclosed properties, it enables a local therapy, thus minimizing the likelihood of systemic toxicity. Despite these advantages, the delivery of siRNA to the posterior segment of the eye, and to the RPE in particular, is not trivial for several reasons. Naked siRNA is prone to nuclease degradation, and it is too large, hydrophilic, and negatively charged to overcome the barriers present in the eye. In addition, siRNA can also stimulate the immune system through the Toll-like receptor pathway [20,21]. These disadvantages can be overcome, to some extent, by incorporating the siRNA into a carrier, leading to the formation of siRNA/carrier complexes. The suitability of a certain carrier varies depending on the tissue, thus careful in vitro testing in physiologically relevant tissue models is required to select the optimal siRNA delivery system for the tissue of interest. In the case of the RPE, various carriers have been evaluated for their siRNA knockdown efficacy in vitro [22,23,24,25,26]; unfortunately, these studies have been performed using dividing cells that do not appropriately represent the terminally differentiated, post-mitotic RPE found in vivo [27]. For example, our recent study [28] has shown that the promising knockdown levels obtained with dividing RPE cells did not translate to adequate efficacy in differentiated cells. Similar trends have also been observed with the transfection efficiency of plasmid DNA in several cell lines [29,30,31].

This study examines the suitability of a polymer carrier (poly(benzyl-l-glutamate) and poly-l-lysine), a lipid carrier (1,2-Dioleoyl-3-trimethylammoniumpropane (DOTAP), 1,2-dioleoyl-sn-glycero-3-phosphoethanolamine (DOPE), and protamine sulfate (PS) liposomes), a lipidoid carrier (1,2-distearoyl-sn-glycero-3-phosphoethanolamine-N-[amino(polyethylene glycol)-2000] (DSPE-PEG), cholesterol, lipidoid), and a few commercial carriers as siRNA delivery vehicles. The knockdown efficacy is evaluated at the mRNA and at protein level in relevant RPE cell models (human ARPE-19 cell line, primary porcine RPE, and human embryonic stem cell-derived RPE), at different maturation stages. The effect of cell differentiation, including the role of melanin, on the siRNA knockdown efficiency is assessed.

## 2. Materials and Methods

### 2.1. Materials

Chloroform (CH_2_Cl_2_), methanol (MeOH), ammonium hydroxide (NH_4_OH), spermidine, 1,2-epoxytetradecane (ETD), 2-(N-morpholino)ethanesulfonic acid (MES), 4-(2-hydroxyethyl)piperazine-1-ethanesulfonic acid (HEPES), protamine sulfate (PS), chloroform, collagen IV from human placenta, GAPDH siRNA (SS: 5′-GGUCAUCCAUGACAACUUU[dT][dT]-3′, AS: 5′-AAAGUUGUCAUGGAUGACC[dT][dT]-3′), HPRT1 siRNA (SS: 5′-CCAGUAAAGUUAUCACAUGUUCUdAdG-3′, AS:5′-CUAGAACAUGmUGmAUAACUUUmACmUGmGmUmG-3′), control siRNA (SS: 5′-UUCUCCGAACGUGUCACGUTT-3′, AS: 5′-ACGUGACACGUUCGGAGAATT-3′), unlabeled DNA strand (5′-ACTTGTGGCCGTTTACGTCGC-3′), human GAPDH primers (F: 5′-GTCAGCCGCATCTTCTTTTG-3′, R: 5′-GCGCCCAATACGACCAAATC-3′), human β-actin primers (F: 5′-AGAGCTACGAGCTGCCTGAC-3′, R: 5′-AGCACTGTGTTGGCGTACAG-3′), porcine HPRT1 primers (F: 5′-GGTCAAGCAGCATAA-3′, R: 5′-GGCATAGCCTACCAC-3′), and porcine GAPDH primers (F: 5′-GATCATCAGCAATGCCTCCT-3′, R: 5′-TGTGGTCATGAGTCCTTCCA-3′) were purchased from Sigma-Aldrich (St. Louis, MO, USA). The matrigel growth factor reduced (GFR) basement membrane matrix was from Corning Inc. (Corning, NY, USA). IL-6 siRNA (SS: 5′-GAACGAAUUGACAAACAAAtt-3′, AS: 5′-UUUGUUUGUCAAUUCGUUCgt-3′), carboxyfluorescein-labeled IL-6 siRNA (FAM-IL-6 siRNA) with the same sequence, and Atto565-labeled DNA strand (5′-Atto565-GACGTAAACGGCCACAAGTTC-3′) were purchased from Eurofins Genomics (Ebersberg, Germany). Nuclease-free water, Alexa Fluor 488 TFP ester, and all the cell culture reagents (unless stated otherwise) were from Thermo Fisher Scientific (Walthan, MA, USA). 1,2-distearoyl-sn-glycero-3-phosphoethanolamine-N-[amino(polyethylene glycol)-2000] (DSPE-PEG), 1,2-Dioleoyl-3-trimethylammoniumpropane (DOTAP), and 1,2-dioleoyl-sn-glycero-3-phosphoethanolamine (DOPE) were from Avanti Polar Lipids (Alabaster, AL, USA). The lipids were used without further purification. The extruder was also from Avanti Polar Lipids and the 100 nm Nuclepore^®^ polycarbonate membranes (diameter 19 mm) were from Whatman Int. Ltd. (Maidstone, UK).

### 2.2. siRNA Delivery Systems

#### 2.2.1. Polyplexes

Cationic micelles were prepared by dissolving the amphiphilic polypeptide block copolymer PBE_30_-b-PK_30_ [32] in nuclease-free water at a stock concentration of 1 mg/mL. Then, the siRNA and the micellar solution were both diluted in 10 mM HEPES (pH 7.2), mixed together, and incubated at room temperature for at least 20 min to form polyplexes. The used charge ratio (N/P) was 8/1. The charge ratio was the molar ratio of positive/negative charges originating from the amines of the carrier and the phosphate of the siRNA strand, respectively (N/P).

#### 2.2.2. Lipoplexes

Cationic liposomes composed of DOTAP and DOPE were prepared by the thin lipid hydration method described by Ruponen et al. [33]. Shortly, the lipids were mixed at a molar ratio of 1:1 (3.2 mM) in chloroform. The organic solvent was evaporated to obtain a lipid film, which was hydrated in nuclease-free water at +30 °C for 30 min. After 2 h at room temperature, the liposome solution was sonicated for 5 min at +30 °C and extruded several times through a 100 nm polycarbonate membrane. The cationic liposomes were stored at +4 °C and used within three months of preparation.

The DOTAP/DOPE/PS-siRNA lipoplexes were prepared by mixing 4.7 µg of siRNA with 10 µg of protamine sulfate solution in MES-HEPES (pH 7.2). After 10 min of incubation, the DOTAP/DOPE liposomes were added. The lipoplexes were incubated for 20 min. The charge ratio (N/P) of the formed lipoplexes was 4.5/1 [34].

#### 2.2.3. Lipidoid-siRNA Complexes

Lipid-like molecules (lipidoid) were synthesized by a ring-opening reaction between 1,2-epoxytetradecane (ETD) and the amines as previously described [35]. Briefly, spermidine and ETD at a molar ratio of 1:4 were added into a 2 mL glass vial, then the mixture was stirred at 90 °C and protected from light for 2 days. The product was purified by silica gel column chromatography with a gradient elution from CH_2_Cl_2_ to 75:22:3 CH_2_Cl_2_/MeOH/NH_4_OH. The transparent, pale yellow, oily product was stored at −20 °C.

DSPE-PEG/lipidoid/cholesterol nanoparticles were prepared via the nanoprecipitation method and purified by centrifugation [36]. Briefly, the synthesized lipidoid, cholesterol, and DSPE-PEG were dissolved in 100% ethanol at concentrations of 100, 25 and 100 mg/mL, respectively, and combined at a weight ratio of 2:2:1. The mixture was then injected quickly into acetate buffer (200 mM, pH 5.4) while stirring to form empty nanoparticles. The DSPE-PEG/lipidoid/cholesterol-siRNA complexes (briefly, lipidoid-siRNA complexes) were formed by mixing 10 pmol of siRNA (for a 50 nM siRNA dose) with the empty nanoparticles in nuclease-free water at an RNA to nanoparticle ratio of 1 µg to 7.5 µL. After gentle mixing with a pipette, the solutions were incubated for 20 min at room temperature. The lipidoid-siRNA complexes were used immediately.

#### 2.2.4. Commercial Carriers

Lipofectamine 2000 (Thermo Fisher Scientific, Walthan, MA, USA), DharmaFECT 4 (Dharmacon, Horizon Discovery, Cambridge, UK), and Metafectene PRO (Biontex, Munich, Germany) were used as positive controls according to the manufacturers’ instructions. The weight (μg) RNA to volume (μL) carrier ratio was 1:3.3 for Lipofectamine 2000 (L2000), 1:3 for DharmaFECT 4 (DF4), and 1:5 for Metafectene PRO (PRO). After gentle mixing with a pipette, the solutions were incubated for 20 min at room temperature to allow the formation of nanoparticles. The commercial complexes were used immediately.

### 2.3. Dynamic Light Scattering (DLS) and Nanoparticle Tracking Analysis (NTA)

The hydrodynamic diameter of the siRNA complexes was determined by DLS, using a Zetasizer APS (Malvern Panalytical, Spectris plc, Egham, Surrey, UK) with a nominal 5 mV He-Ne laser operating at a 633 nm wavelength. The refractive index of the lipid-based carriers and polymer-based carriers was 1.33 and 1.450, respectively, and the viscosity was 0.8872 cP at 25 °C. For each sample, three separate measurements were conducted with 20 runs each. The size of the lipidoid and commercial complexes was analyzed by NTA using a NanoSight LM10 system (Malvern Panalytical, Spectris plc, Egham, Surrey, UK) fitted with a high-sensitivity cCMOS camera and a 405 nm laser. Each sample was appropriately diluted in PBS before the measurement. Videos of 60 s were recorded and analyzed with the NTA software (version 3.1, build 3.1.45). The concentration ranges of the particles were between 10^8^ and 10^9^ particles per mL. Three measurements per sample were taken.

### 2.4. Retinal Pigment Epithelium (RPE) Cell Cultures

#### 2.4.1. Human Retinal Pigment Epithelial Cells (ARPE-19)

ARPE-19 cells from ATCC (Manassas, VA, USA) were cultured in a growth medium composed of Dulbecco’s modified Eagle’s medium/Nutrient Mixture F12, supplemented with 10% fetal bovine serum (FBS), 2 mM of L-glutamine, 50 U/mL of streptomycin, and 50 U/mL of penicillin. The cells were maintained at +37 °C in 7% CO_2_, sub-cultured (1:3 or 1:5) once a week, and the medium was renewed once a week. The ARPE-19 cells were differentiated, as previously described, on laminin-coated Costar Transwell inserts (pore size 0.4 µm, Corning, NY, USA) at a seeding density of 160,000 cells/cm^2^ [34]. The growth medium was renewed twice a week. The composition of the growth medium was the same as above, except the FBS concentration was 1%. The cells were grown on inserts for 3–4 weeks. The tight junction formation was evaluated by a transepithelial electrical resistance measurement (TEER).

#### 2.4.2. Primary Porcine RPE (pRPE)

Fresh porcine eyes were obtained from a local slaughterhouse (Jensens Slagtehus, Hadsten, Denmark). The porcine eyes were kept on ice during transport and handled immediately upon arrival to the laboratory. The extraocular tissues were removed with curved iris scissors. The anterior part of the eye, the lens and the vitreous, were gently removed. The eye cup was placed in a well plate and filled with PBS. After 10 min, the PBS was removed, together with the neural retina. The eye cup was then filled with 0.25% trypsin-EDTA and placed in the incubator at +37 °C for 30 min. After this time, the detached RPE cells were collected. The RPE cells of 20 eyes were collected in 50 mL tubes containing 1 part trypsin-EDTA with the cells and 2 parts cell culture medium (low glucose DMEM supplemented with pyruvate, 10% FBS, 2 mM GlutaMAX, and 1% penicillin-streptomycin (10,000 U/mL)). The tubes were centrifuged three times at 1000 rpm for 2 min, and each time the supernatant was replaced by fresh culture medium. Then, the cells were placed in a T25 tissue culture flask and the medium was replaced twice a week until the cell monolayer reached confluency. The pRPE cells were then seeded on collagen IV-coated (11.2 µg collagen/insert) Transwell inserts (12 well) at a density of 160,000 cells/cm^2^ and cultured for 4 weeks. The cell culture medium had the same composition as described above, except it contained only 1% FBS. The tight junction formation was evaluated by a transepithelial electrical resistance measurement (TEER).

#### 2.4.3. Human Embryonic Stem Cell-Derived RPE (hESC-RPE)

The pluripotent hESC line Regea15/025 (46,XX) was derived [37], characterized, and maintained on inactivated human foreskin fibroblasts (hFF) (CRL-2429, ATCC, Manassas, VA, USA) in serum-free hESC culture medium, as previously described [38]. The spontaneous differentiation of RPE from pluripotent hESCs in floating aggregates using the RPEbasic differentiation method and subsequent selection and enrichment were performed as previously described [38,39]. The hESC-RPE cells were matured on Matrigel-coated inserts (13.44 µg Matrigel/insert, Millicell Hanging Inserts, MCRP24H48, seeding density 75,000 cells/insert) for 2 and 4 weeks. The tight junction formation was evaluated by a transepithelial electrical resistance measurement (TEER).

### 2.5. Transepithelial Electrical Resistance Measurement (TEER)

The TEER of the cell monolayers was measured with a Millicell ERS-2 Volt-Ohm Meter equipped with a silver/silver chloride electrode (EMD Millipore Corporation, Billerica, MA, USA). Briefly, the STX01 electrode was sanitized with 70% ethanol for 15 min immediately before use, then the electrode was used to carry out the measurements. To obtain the actual monolayer resistance, the resistance reading across a blank insert (insert without cells) was subtracted from the resistance reading across the monolayer (insert with cells). The unit area resistance (Ω cm^2^) was calculated by multiplying the resistance reading (Ω) with the effective surface area of the filter membrane (cm^2^).

### 2.6. Cytotoxicity

The cytotoxicity of the polyplexes, lipoplexes, and L2000/siRNA complexes was determined by an MTT assay. The ARPE-19 cells were seeded on a 24-well plate at a density of 100,000 cells/well in 500 µL of supplemented growth medium. The next day, the cells were washed with PBS (1x, pH 7.2). The formulations with 50 nM of siRNA were incubated with the cells in Opti-MEM for 5 h. Then, the cells were washed with PBS and incubated for 4 h with 400 µL of 0.5 mg/mL thiazolyl blue tetrazolium bromide (MTT, Sigma-Aldrich, St. Louis, MO, USA). After incubation, formazan crystals of the living cells were solubilized by adding 100 µL of 10% sodium dodecyl sulphate −0.1 M hydrochloric acid. On the following day, the formazan amount was quantified by measuring the absorbance at 570 nm (Varioskan Flash, Thermo Fisher Scientific, Waltham, MA, USA). The cell viability was calculated as a percentage of the non-treated cells.

The proliferation of the cells transfected with the lipidoid and DF4 complexes was measured by an alamarBlue assay following the instructions given by the manufacturer. This assay was used with differentiated cells, because it did not require cell lysis. Briefly, an amount equal to 10% of cell culture volume (50 µL) of alamarBlue reagent was added apically and the cells were returned to the incubator for one hour. Thereafter, 100 µL of cell culture medium was transferred to a black well plate, and the fluorescence was measured at λ_ex_ 560 nm and λ_em_ 590 nm with a plate reader (Varioskan Flash, Thermo Fisher Scientific, Waltham, MA, USA). The cell proliferation was calculated relative to the non-treated cells.

### 2.7. Gene Silencing

#### 2.7.1. Interleukin 6 (IL-6) Protein Knockdown

Dividing cells: The ARPE-19 cells were seeded on a 24-well plate at a density of 100,000 cells/well in 500 µL of supplemented growth medium. Next day, the cells were transfected with 50 nM of IL-6 siRNA in Opti-MEM for 5 h. After transfection, the cells were washed with PBS, and 1 mL of growth medium containing 10 μg/mL of lipopolysaccharide (LPS, Escherichia coli O55:B5 Sigma-Aldrich, St. Louis, MO, USA) was added to the cells. The LPS was used to induce IL-6 secretion [40]. Samples of 500 µL were collected from the supernatant 72 h after the polyplex and lipoplex removal and replaced with the equal volume of LPS medium. The amount of IL-6 was analyzed from the collected medium samples using a human IL-6 ELISA kit according to the instructions provided by the manufacturer (Gen-Probe, Diaclone SAS, Besançon, France). The relative IL-6 protein secretion was calculated by comparing the exposed cells to the LPS-treated cells without any exposure (100%).

Differentiated cells: The ARPE-19 cells cultured for 3–4 weeks were transfected apically with 100 nM of IL-6 siRNA in Opti-MEM for 5 h. After incubation, the medium was removed and the cells were washed with PBS. Growth medium supplemented with 10 μg/mL of LPS was added to the apical and basolateral sides of the cells. Medium samples were collected at 2, 5, and 9 days. The removed samples were replaced with an equal volume of the LPS medium. The samples were analyzed with the human ELISA kit described above.

#### 2.7.2. Glyceraldehyde-3-Phosphate Dehydrogenase (GAPDH) and Hypoxanthine Phosphoribosyltransferase 1 (HPRT1) MRNA Knockdown

The differentiated cells were transfected apically with 10, 50, or 100 nM of GAPDH siRNA (ARPE-19 and hESC-RPE) or HPRT1 siRNA (pRPE) overnight in full medium, including serum. Two days after transfection, the total RNA was isolated using the NucleoZOL reagent (Macherey-Nagel, Düren, Germany). The total RNA concentration was determined with a DeNovix DS-11 spectrophotometer (DeNovix Inc., Wilmington, DE, USA). The cDNA was synthesized using the RevertAid First strand cDNA synthesis kit (Thermo Fisher Scientific, Waltham, MA, USA). The quantification of the GAPDH mRNA, HPRT1 mRNA, and β-actin mRNA was performed with the LightCycler 480 SYBR Green I Master (Roche, Basel, Switzerland). The primer concentration was 250 nM, and 4 ng of cDNA was used in the final 10 µL reaction volume. All the quantitative PCRs were performed on a LightCycler^®^ 480 Instrument (Roche, Basel, Switzerland). The sequence-specific amplification of cDNAs was verified by performing a melting-point analysis. For the differentiated ARPE-19 and hESC-RPE cells, the GAPDH mRNA expression was normalized to the endogenous β-actin, whereas for the pRPE cells, the HPRT1 mRNA expression was normalized to the endogenous GAPDH.

### 2.8. Cell Uptake

The cells were transfected as in paragraph 2.7 using either FAM-IL-6 siRNA or Alexa Fluor 488 dsDNA (21-mer). After incubation, the cells were washed three times with Hanks’ Balanced Salt Solution (HBSS) (Thermo Fisher Scientific, Waltham, MA, USA) supplemented with 10 mM of HEPES (pH 7.4). Then, the cells were detached with trypsin, re-suspended in the HBSS-HEPES buffer, and centrifuged at 1200 rpm for 10 min. The cells were analyzed by flow cytometry (LSR II, Becton, Dickinson and Company, Franklin Lakes, NJ, USA and Gallios, Beckman Coulter Life Sciences, Indianapolis, IN, USA). The data collection and analysis were controlled by the FACSDiva software (BD Biosciences) and Kaluza software (Beckman Coulter Life Sciences), where 10,000 cell events were collected for each sample. Non-treated and naked FAM-IL-6 siRNA-treated cells were used for setting the baseline and for calculating the relative cell association of the siRNA formulations.

### 2.9. Intracellular Distribution

The intracellular distribution of Atto565 dsDNA (21-mer) was observed with confocal microscopy. The hESC-RPE cells were transfected as in paragraph 2.7. The lysosomes were labeled with the CellLight Lysosomes-GFP reagent (Thermo Fisher Scientific, Waltham, MA, USA) following the instructions given by the manufacturer. Briefly, 24 h after transfection, the CellLight reagent (16 µL) was added and incubated with the cells for ≥ 16 h. Thereafter, the cells were fixed with 4% paraformaldehyde for 10 min at room temperature and mounted in Vectashield mounting medium (Vector Laboratories Inc., Burlingame, CA, USA). The melanosomes were labeled by immunocytochemistry as the following. The cells were fixed with 4% paraformaldehyde for 10 min at room temperature, followed by permeabilization with 0.5% saponin in PBS for 10 min and blocking with 1% bovine serum albumin (BSA), 22.52 mg/mL glycine in PBST (PBS +0.1% Tween 20) for 30 min. The cells were then incubated with the primary polyclonal antibody Rab27a (1:200, Sicgen, Cantanhede, Portugal) in 1% BSA in PBST in a humidified chamber for 1 h at room temperature, followed by incubation with the secondary antibody (donkey anti-goat Alexa Fluor 488, 1:500, Thermo Fisher Scientific, Waltham, MA, USA) in 1% BSA in PBST for 1 h at room temperature. The cell nuclei were stained with the DAPI included in the Vectashield mounting medium. Images were captured on an Olympus IX-83 fluorescent microscope with the Andor confocal spinning unit and Andor iXon Ultra 897 camera, Olympus Upsalo W, 60x/1.20 NA water objective lens, using the Olympus cellSens software (Olympus). Multichannel images were processed using the Fiji software (ImageJ 1.52s).

### 2.10. Statistical Analysis

A one-way ANOVA followed by Tukey’s multiple comparisons test was performed using GraphPad Prism version 8.4.2 for macOS (GraphPad Software, San Diego, CA, USA). The significance level of the test was set to 0.05.

## 3. Results

### 3.1. Particle Size and Polydispersity Index (PdI)

The particle size and polydispersity index (PdI) of the siRNA/carrier complexes, evaluated by DLS and NTA, are summarized in Table 1. The PBE_30_-b-PK_30_ polyplexes were around 50 nm and had a narrow size distribution, whereas the DOTAP/DOPE/PS lipoplexes were about 200 nm in diameter and displayed a higher polydispersity. The lipidoid-siRNA complexes (lipidoid) were approximately 150 nm in diameter and had a monodisperse distribution. Comparable results were obtained when their sizes were measured with NTA. The commercial complexes were measured by NTA, since the phenol red present in the nanoparticle preparation medium disturbed the DLS measurement. The commercial complexes sizes were 160–280 nm and polydisperse.

### 3.2. Cytotoxicity

The in vitro toxicity of all the siRNA/carrier complexes was low, and the RPE cells maintained an over 80% viability after treatment (Figure 1).

### 3.3. Knockdown Efficacy in RPE Cell Models

#### 3.3.1. Human ARPE-19

Dividing ARPE-19 cells: The IL-6 protein knockdown was about 70% (*p* < 0.0001) with the lipoplexes (Figure 2), whereas no knockdown was measured with the polyplexes, even after treatment with chloroquine, a commonly used lysosomotropic agent that promotes endosomal escape and enhances transfection [41] (Appendix A). L2000 resulted in a 60% (*p* = 0.0003) decrease in protein secretion. There was no statistically significant difference between the knockdown efficacy of the lipoplexes and L2000 (*p* = 0.4663). The negative control siRNA did not promote silencing.

Differentiated ARPE-19 cells: The IL-6 protein knockdown was highest with the lipoplexes, at about 60% (*p* < 0.0001) 5 days after transfection (Figure 3), whereas the L2000 resulted in a 40% (*p* = 0.0207) decrease in protein secretion at the same time point. The knockdown efficacy between the lipoplexes and L2000 at day 5 was not statistically significant (*p* = 0.5235). Interestingly, the silencing effect of lipoplexes was delayed compared to L2000, and prolonged to at least 9 days after transfection (*p* < 0.0001). The polyplexes were not tested with the differentiated ARPE-19 cells, since no knockdown was detected in the dividing cells. The negative control siRNA did not promote silencing. The siRNA dose was doubled (100 nM) in the differentiated cells, because no knockdown could be measured with the same dose used in the dividing cells (50 nM). The TEER of the ARPE-19 cells was 39 ± 5 Ω cm^2^. Figure 3 illustrates the decrease in IL-6 secretion on the apical side of the monolayer; the basolateral IL-6 secretion was also measured with similar results (data not shown).

The lipidoid-siRNA complexes-mediated GAPDH mRNA silencing in differentiated ARPE-19 cells (Figure 4) was achieved with the same 50 nM dose used in the dividing cells. After transfection, the GAPDH gene expression was decreased by 60% (*p* = 0.0019). The DF4-mediated silencing was approximately 90% (*p* < 0.0001), however there was no statistical significance between the lipidoids and DF4 (*p* = 0.1530). The lipidoids retained a 64% gene silencing efficacy even at a low 10 nM siRNA dose (data not shown). The negative control siRNA did not promote silencing. The TEER of the ARPE-19 cells in this experiment was 78 ± 13 Ω cm^2^.

#### 3.3.2. Primary Porcine RPE (pRPE)

A siRNA dose of 50 nM was sufficient to silence the housekeeping gene HPRT1 in differentiated pRPE cells with lipidoids (Figure 5). After transfection, the remaining HPRT1 gene expression was 45% (*p* = 0.0294). The DF4 achieved a 50% knockdown (*p* = 0.0482). The negative control siRNA did not promote silencing. The TEER of the pRPE cells was 853 ± 39 Ω cm^2^.

#### 3.3.3. Human Embryonic Stem Cell-derived RPE (hESC-RPE)

A siRNA dose of 50 nM in lipidoids silenced the GAPDH by 50% (*p* = 0.0004) in 2-week matured hESC-RPE cells (Figure 6a). The best carrier was PRO, with a knockdown efficiency of 75% (*p* < 0.0001), while the efficiency of the other commercial carrier DF4 was 55% (*p* < 0.0001). The negative control siRNA did not promote statistically significant silencing for PRO and lipidoids. The TEER of the 2-week matured hESC-RPE cells was 18 ± 5 Ω cm^2^. 

Despite the promising results obtained with the 2-week matured hESC-RPE cells, no statistically significant knockdown was achieved in the 4-week matured cells (Figure 6b). The negative control siRNA did not promote silencing. The TEER of the 4-week matured hESC-RPE cells was 435 ± 47 Ω cm^2^. The 2-week matured cells were not pigmented, whereas pigmentation is clearly visible in the 4-week matured cells.

### 3.4. Cell Uptake

Cell uptake studies were performed in order to evaluate the effect of cell differentiation on the endocytosis of siRNA complexes. The carriers tested in the dividing ARPE-19 cells (polyplexes, lipoplexes, and commercial carriers) all had a 100% cell association (measured as the % of positive cells, Figure 7a). In the differentiated ARPE-19 and 4-week matured hESC-RPE cells instead, the lipidoids had the highest % of positive cells (>90% in hESC-RPE, Figure 7b), while the lipoplexes had 20% positive ARPE-19 cells. The L2000 and DF4 had 20% (ARPE-19) and 40% (hESC-RPE) positive cells, respectively.

### 3.5. Intracellular Distribution in hESC-RPE

The intracellular distribution of the fluorescently labelled dsDNA (used as a non-biologically active mimic for siRNA) was studied in order to explain the striking knockdown differences observed in the hESC-RPE that were matured for 2 and 4 weeks (Figure 6a,b).

#### 3.5.1. 2-Week Matured hESC-RPE

The intracellular distribution of the fluorescently labelled dsDNA (used as a non-biologically active mimic for siRNA, in red) was dependent on the type of carrier. For DF4 (Figure 8a) and especially for PRO (Figure 8b), cytosolic staining (arrowheads), which may indicate endosomal release, was observed. With these carriers, the labelled dsDNA also localized inside large aggregates (arrows). In the case of lipidoids (Figure 8c), the dsDNA was found inside discrete, small vesicles, which did not colocalize with lysosomes (in green). The Z-stack images indicate that for all the tested carriers, dsDNA was found inside the cells.

#### 3.5.2. 4-Week Matured hESC-RPE

PRO did not distribute evenly in the cytoplasm as in the 2-week matured cells (Figure 8b); rather, it accumulated inside fairly large apical vesicles (Figure 9a and Appendix A). The lipidoids localized on the apical cell membrane (Figure 9b and Appendix A) but were not found inside cells. Subsequent staining with melanosomal marker Rab27a revealed that PRO was localized inside melanosomes (Figure 9c–e) on the apical side of the cell monolayer.

## 4. Discussion

The pathogenesis of several age-related ocular disorders is characterized by the presence of continuous, low-grade inflammation. Silencing pro-inflammatory genes by siRNA is an attractive therapeutic option to reduce inflammation and restore homeostasis in the retina. Despite the undeniable potential of siRNA therapeutics, a key obstacle still hampering their widespread use is the safe and efficient delivery to target cells. Gene silencing in the RPE requires a carrier that can successfully deliver the siRNA to a quiescent, terminally differentiated cell monolayer. siRNA therapeutics must first of all gain access into cells, a process which occurs via endocytosis and whose mechanisms have been widely studied. It is worth keeping in mind, however, that most of our current knowledge on endocytic mechanisms is derived from studies of proliferating cells, and that the mechanisms prevailing in non-dividing cells might be quite different [42]. Several studies have shown, for example, that cell differentiation causes a reduction in the uptake of different cargoes in intestinal [43,44,45], kidney [46], and airway [47] epithelial cells. While we could not find any published literature on the effect of cell differentiation on the endocytic activity of RPE cells, we also observed a decreased uptake of siRNA complexes in differentiated ARPE-19 cells compared to dividing cells. While lipidoid-siRNA complexes did retain a > 90% cell association in 4-week matured hESC-RPE cells, a confocal microscope analysis revealed that they were not internalized inside cells, but instead they localized apically at the cell surface. Thus, our results indicate that, similarly to other epithelial tissues [43,44,45,46,47], differentiation alters the endocytic activity of RPE cells and causes a decrease in the uptake of siRNA complexes.

RPE cell differentiation also lessened the efficacy of gene silencing. While clearly a less efficient cell uptake may partly explain the decreased knockdown, an additional important barrier for efficient siRNA therapy was revealed to be the sequestration of dsDNA (used as siRNA mimic) by melanosomes. Melanosomes do not normally occur in ARPE-19 cells [48] and 2-week matured hESC-RPE (Figure 6a), whereas the 4-week matured hESC-RPE were heavily pigmented (Figure 6b). Some drugs have long been known to be stored inside melanosomes via melanin binding. Indeed, the first reports describing the accumulation of drugs inside ocular melanin were published more than 60 years ago [49,50,51], and several thousands of compounds, mostly small molecular weight drugs, have been investigated for melanin binding in vitro and in vivo since [52,53]. Very few studies, however, describe the melanin binding of oligonucleotides and nanoparticles. Pitkänen and colleagues [54] found that oligonucleotides did not bind to isolated and synthetic melanin, whereas Geng and coworkers [55] showed that melanin from bacteria interacted with DNA. Despite these conflicting results, the entrapment of siRNA complexes inside melanosomes may not require melanin binding. Schraermeyer and colleagues, for example, observed the accumulation of gold nanoparticles (10–20 nm) used to label rod outer segments [56] and latex particles (100 nm) [57] in the narrow space between melanin and the melanosome membrane in the RPE of bovine RPE-choroid tissue explants. Melanosomes, whose membranes and contents are largely derived from endosomes, are members of the lysosome-related organelle (LRO) family [58]. Previous work has found that, while the vast majority of siRNA lipoplexes that enter cells end up in endolysosomes, only a few percent of siRNA reach the cytosol, either by direct fusion between the lipoplexes and the plasma membrane [59] or via endosomal escape [60,61]. Accordingly, only a fraction of the siRNA dose that localizes in the cytosol is responsible for the gene silencing effect. In our study, we observed the significant cytoplasmic distribution of siRNA in 2-week matured hESC-RPE (Figure 8b). In addition, none of the different complexes co-localized with lysosomes. Upon cell pigmentation at 4 weeks, however, the cytoplasmic staining disappeared, and, in the case of Metafectene PRO, the siRNA localized inside melanosomes instead. This also corresponded to a loss in gene silencing. It is possible that if, upon delivery to the melanosomes, the siRNA is still incorporated into lipoplexes, then the cationic lipoplexes may bind to the polyanionic melanin [53]. This binding may prevent the interaction of lipoplexes with the melanosomal membrane, and thus preclude the release of siRNA into the cytosol.

In summary, our study highlights the importance of picking a physiologically relevant RPE cell model for the selection of siRNA delivery systems. While it is tempting to evaluate their efficacy only in dividing, non-pigmented RPE cells, there is a real risk that these systems will not work in vivo. Given their easy culture, low costs, and quick growth, dividing ARPE-19 cells are still valuable in the early selection rounds and when large preliminary screens are performed. Nevertheless, advanced selections should be carried out with RPE models that better mimic the outer blood-retina barrier—that is, differentiated, pigmented RPE cells. This simple but important choice may lead to a more stringent selection of delivery systems and increase their chance of success in vivo.

## Figures and Tables

**Figure 1 pharmaceutics-12-00667-f001:**
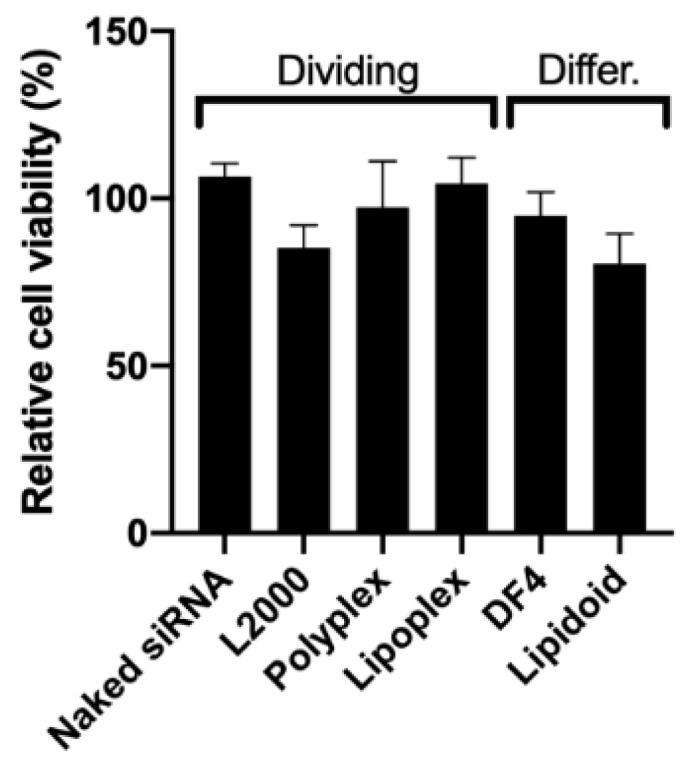
Viability of the dividing and differentiated ARPE-19 after exposure to siRNA/carrier complexes at a siRNA dose of 50 nM. Cell viability was evaluated by MTT (in dividing cells) and an alamarBlue assay (in differentiated cells) after 5 h of exposure to the complexes, and is presented as the percentage of untreated cells. Data are presented as mean + SD, *n* = 3.

**Figure 2 pharmaceutics-12-00667-f002:**
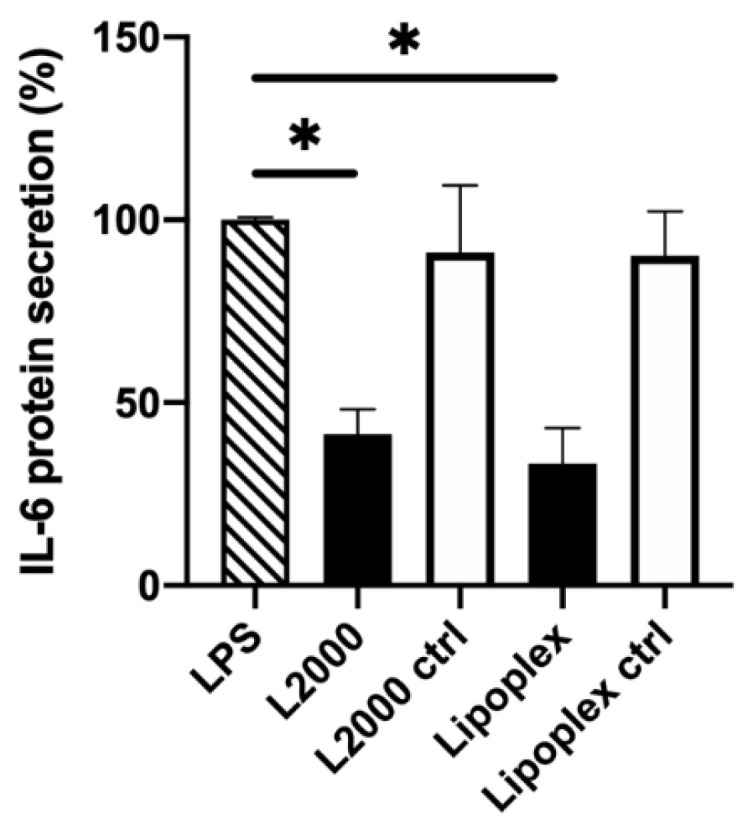
IL-6 knockdown in dividing ARPE-19 cells, 3 days after transfection. Black columns indicate the IL-6 siRNA (50 nM), the white columns indicate the negative control siRNA (50 nM). IL-6 protein secretion was evaluated with ELISA and was normalized to LPS-treated cells (100%, column with diagonal pattern). The basal IL-6 secretion of dividing ARPE-19 cells was 5% relative to the LPS-treated cells. L2000: Lipofectamine 2000; ctrl: negative control siRNA; ***** indicates statistical significance (*p* < 0.05). Data are presented as mean + SD, *n* = 3–4.

**Figure 3 pharmaceutics-12-00667-f003:**
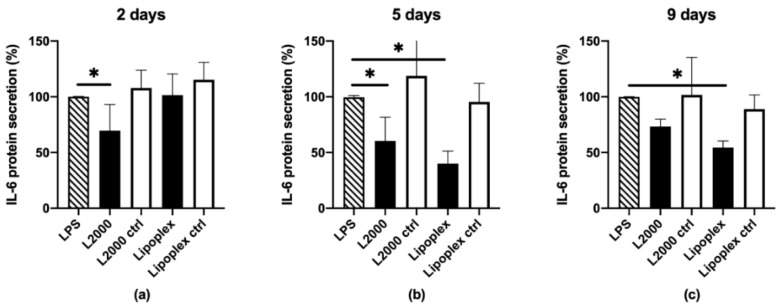
Apical IL-6 protein knockdown in differentiated ARPE-19 cells, 2 (**a**), 5 (**b**), and 9 (**c**) days after transfection. Black columns indicate the IL-6 siRNA (100 nM), the white columns indicate the negative control siRNA (100 nM). IL-6 protein secretion was evaluated with ELISA and normalized to the LPS-treated cells (100%, column with diagonal pattern). L2000: Lipofectamine 2000; ctrl: negative control siRNA; ***** indicates statistical significance (*p* < 0.05). Data are presented as mean + SD, *n* = 3–6.

**Figure 4 pharmaceutics-12-00667-f004:**
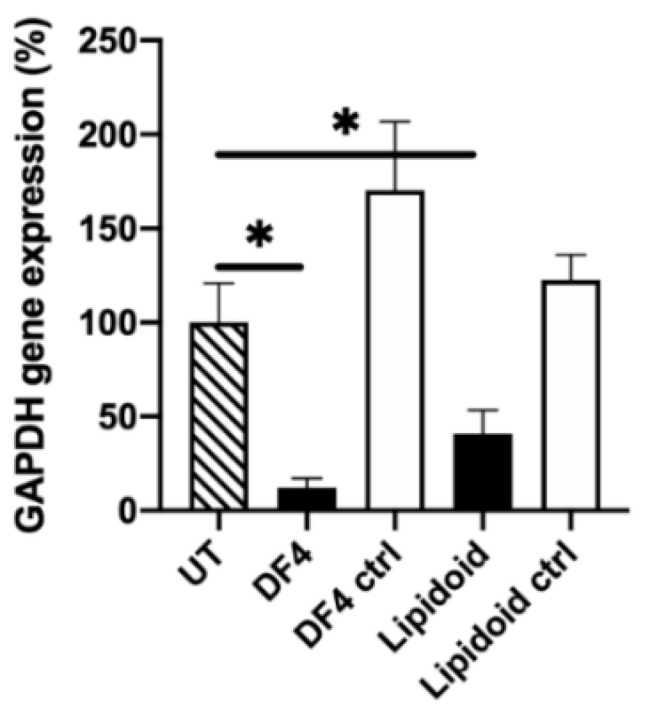
GAPDH mRNA knockdown in differentiated ARPE-19 cells, 2 days after transfection. Black columns indicate the GAPDH siRNA (50 nM), the white columns indicate the negative control siRNA (50 nM). GAPDH mRNA expression was evaluated with a qPCR and normalized to the endogenous β-actin. Results are presented as the % of remaining gene expression. UT: untreated control cells; DF4: DharmaFect 4; ctrl: negative control siRNA; ***** indicates statistical significance (*p* < 0.05). Data are presented as mean + SD, *n* = 3–5.

**Figure 5 pharmaceutics-12-00667-f005:**
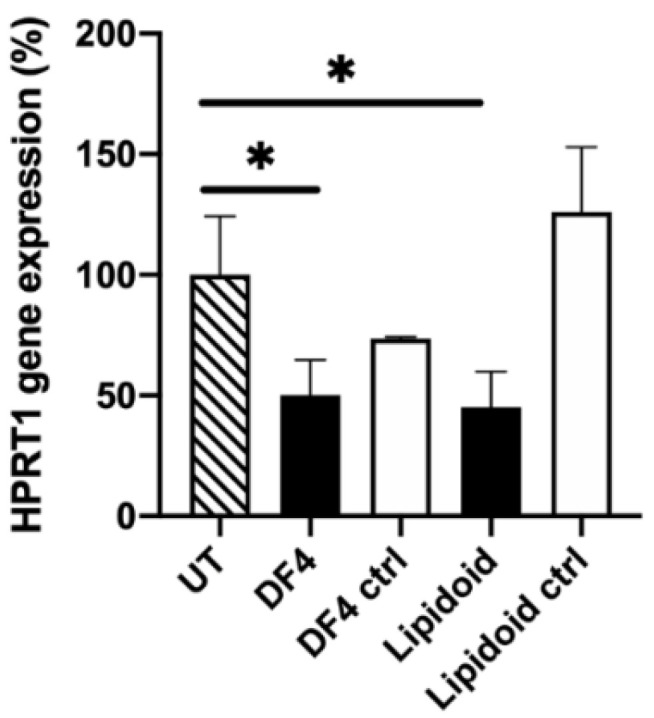
HPRT1 mRNA knockdown in the differentiated primary pRPE cells, 2 days after transfection. Black columns indicate the HPRT1 siRNA (50 nM), the white columns indicate the negative control siRNA (50 nM). HPRT1 mRNA expression was evaluated with a qPCR and normalized to the endogenous GAPDH. Results are presented as the % of remaining gene expression. UT: untreated control cells; DF4: DharmaFect 4; ctrl: negative control siRNA; ***** indicates statistical significance (*p* < 0.05). Data are presented as mean + SD, *n* = 3.

**Figure 6 pharmaceutics-12-00667-f006:**
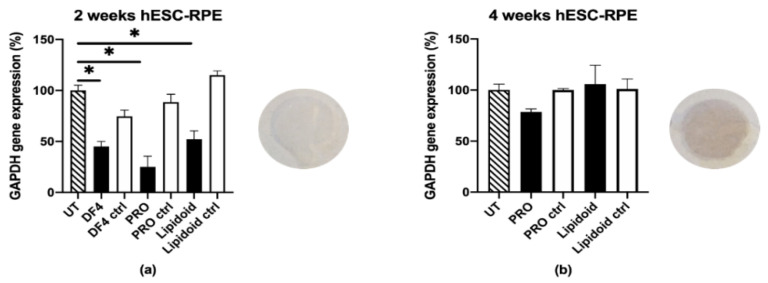
GAPDH mRNA knockdown in (**a**) 2-week and (**b**) 4-week matured hESC-RPE cells, 2 days after transfection. Black columns indicate the GAPDH siRNA (50 nM), the white columns indicate the negative control siRNA (50 nM). GAPDH mRNA expression was evaluated with a qPCR and normalized to the endogenous β-actin. Results are presented as the % of remaining gene expression. UT: untreated control cells; DF4: DharmaFect 4; PRO: Metafectene PRO; ctrl: negative control siRNA; ***** indicates statistical significance (*p* < 0.05). Data are presented as mean + SD, *n* = 3. Cell pigmentation is visible in 4-week matured hESC-RPE.

**Figure 7 pharmaceutics-12-00667-f007:**
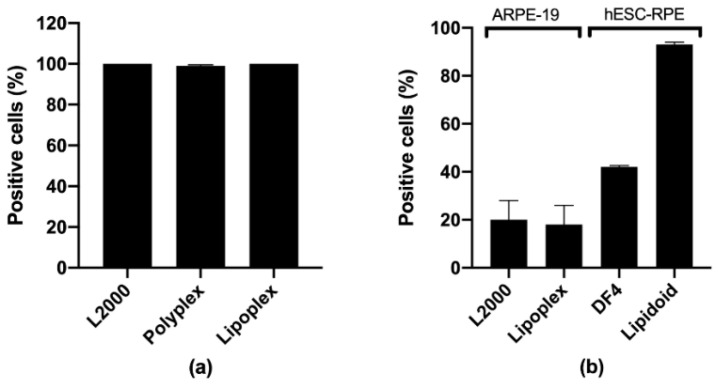
Percentage of positive (**a**) dividing ARPE-19 cells, and (**b**) differentiated RPE cells measured by flow cytometry upon 5 h of treatment with fluorescent siRNA/dsDNA incorporated into different carriers. Dividing ARPE-19 cells were transfected with 50 nM of FAM-IL-6 siRNA, the differentiated ARPE-19 with 100 nM of FAM-IL-6 siRNA, and the 4-week matured hESC-RPE with 50 nM of Alexa Fluor 488 dsDNA (21-mer). L2000: Lipofectamine 2000; DF4: DharmaFect 4. Data are presented as mean + SD, *n* = 3.

**Figure 8 pharmaceutics-12-00667-f008:**
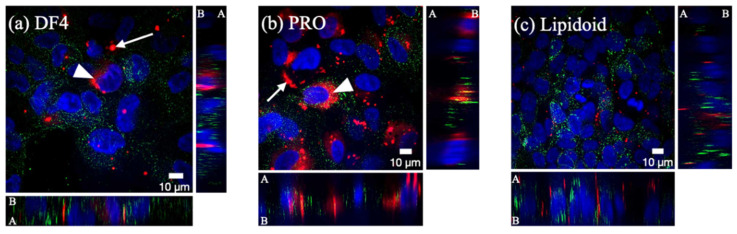
Intracellular distribution of ATTO565-labelled dsDNA (in red) in the 2-week matured hESC-RPE. dsDNA (50 nM) was incorporated into the commercial carriers DF4 (**a**) and Metafectene PRO (**b**), and into lipidoids (**c**). Lysosomes were labelled with CellLight Lysosomes-GFP (in green). Cell nuclei (in blue) were labelled with DAPI. A: apical side; B: basolateral side. Scale bar: 10 μm.

**Figure 9 pharmaceutics-12-00667-f009:**
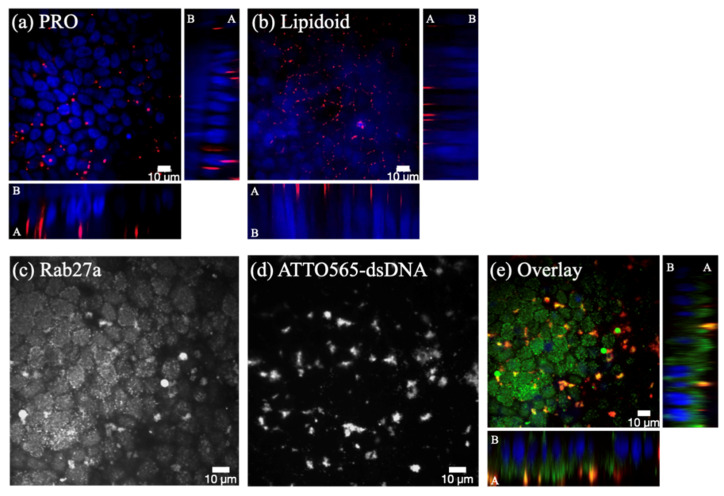
Intracellular distribution of ATTO565-labelled dsDNA (in red) in 4-week matured hESC-RPE. dsDNA (50 nM) was incorporated into the commercial carrier Metafectene PRO (**a**) and into the lipidoids (**b**). (**c**–**e**) Intracellular distribution of Metafectene PRO/dsDNA (50 nM) complexes in 4-week matured hESC-RPE. (**c**) Melanosomes were identified with Rab27a antibody; (**d**) ATTO565-labelled dsDNA; and (**e**) overlay image where melanosomes are in green, dsDNA in red, and cell nuclei in blue (DAPI). A: apical side; B: basolateral side. Scale bar: 10 μm.

**Table 1 pharmaceutics-12-00667-t001:** Size and polydispersity index of siRNA/carrier complexes measured by dynamic light scattering (DLS), and size measured by nanoparticle tracking analysis (NTA).

Carrier	Size, DLS (nm)	PdI	Size, NTA (nm)
Polyplex	46.4 ± 14	0.20	n/a
Lipoplex	216 ± 27	0.40 ± 0.05	n/a
Lipidoid	166 ± 18	0.21 ± 0.03	191 ± 62
DharmaFect 4	n/a	n/a	281 ± 160
Metafectene PRO	n/a	n/a	160 ± 88
Lipofectamine 2000	n/a	n/a	n/a

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
