# Peer review of "Avoiding the Pitfalls of siRNA Delivery to the Retinal Pigment Epithelium with Physiologically Relevant Cell Models"

_pharmaceutics, 2020, doi:10.3390/pharmaceutics12070667_

Round 1

Reviewer 1 Report

Comments and Suggestions for Authors This is an interesting manuscript where the authors put forward her opinion that RPE cell differentiation alters their endocytic activity and causes a decrease in the uptake of siRNA complexes. The study have a different perspective on the ocular gene delivery system. While there are still some questions should be explain. 1.What were the Zeta potentials of polyplexes, lipoplexes, and lipidoid-siRNA complexes. Usually, the cytotoxicity of cationic nanoparticle might be high. Please explain it. 2.In the experiment of Knockdown efficacy, why three different genes were chosen in three RPE cell models respectively? 3.The images of Intracellular distribution in hESC-RPE were not clear and the scale bar should be added. And why only hESC-RPE cell model was used in this experiment?

Reviewer 2 Report

This article by Ramsay et al. shows that siRNA delivery to the retinal pigment epithelium (RPE) is dependent on the cell model chosen. Specifically, they observe a decrease in siRNA uptake and knockdown efficacy when differentiated RPE cells were treated with various siRNA-loaded delivery vehicles as compared to RPE dividing cells. The authors suggest that this difference may explain the discrepancy between in vitro and in vivo siRNA delivery and knockdown in the RPE. They also show preliminary data suggesting that siRNA is sequestered in melanosomes of hESC-RPE, which further supports their argument. The author’s manuscript is thorough, and I would recommend publication in Pharmaceutics pending the following minor revisions:

  • Lines 147-148. Briefly summarize method for making lipidoid-siRNA complexes via nanoprecipitation. Although the method is referenced, it would be helpful if the authors provided a brief synopsis of the protocol for the reader.
  • Figure 1. Explain which data is from the MTT assay vs. alamarBlue assay. I would suggest expanding the figure legend to clarify this and also elaborate on treatment conditions (quantity of siRNA used, timing of treatment, etc.) Although this information is present in the methods section, it would be helpful to have the details summarized in the figure legend to provide clarity. It is also unclear why two different cytotoxicity assays were used for different delivery systems, so I would suggest that the authors explain this decision in the text.
  • Figure 7. In the figure legend, elaborate on methodology used and clarify amount of fluorescent dsDNA transfected, incubation time, analysis technique, etc.
  • Figure 8 and Line 412. I think that more detailed imaging analysis may be required to prove that the regions marked by arrowheads definitively show endosomal release of dsDNA. (For example, see Wittrup et al. in Natura Biotech from 2015; doi: 10.1038/nbt.3298) The authors should revise the statement of their conclusion to convey this uncertainty.
  • Line 457. dsDNA replaced siRNA in experiments assessing melansome sequestration and intracellular distribution, so this should be noted in the conclusion. These assays could have been performed using siRNA, rather than dsDNA, so I am also curious why the authors made this decision.
  • How was siRNA concentration determined after encapsulation in each delivery system, particularly the non-commercial carriers? How was encapsulation efficiency calculated? If encapsulation efficiency was not measured, the authors should add this analysis to their manuscript. This characterization will assure that differences in siRNA-mediated knockdown efficacy are not the result of differences in siRNA concentration due to inconsistent encapsulation efficiencies.
  • Do differences in transepithelial electrical resistance (TEER) values affect interpretation of results? TEER values vary between assays, and it would be helpful if the authors commented on this significance.
  • Do the authors have a suggestion for a suitable delivery system to test in vivo based on this data? This would be a helpful conclusion to add to the discussion.
  • Finally, it would be helpful if the authors explained their experimental decisions in more detail. For example, why were only some delivery systems tested on certain cell types? In Figure 2, IL-6 knockdown is measured in dividing ARPE-19 cells after treatment using L2000 and lipoplexes. The authors mention that polyplexes were not effective, but why were the other delivery systems (DF4, PRO, and lipidoids) not tested? As another example, in Figure 3, IL-6 knockdown is measured in differentiated ARPE-19 cells using L2000 and lipoplexes, but in Figure 4, DF4 and lipidoids are used to assay GAPDH knockdown. Why was the siRNA target changed? The authors should clarify their decisions or state why data is not shown.

Minor comments:

  • Citations needed for lines 56-63.
  • Define “ETD” in Line 142.

Reviewer 3 Report

In the paper "Avoiding the pitfalls of siRNA delivery to the retinal pigment epithelium with physiologically-relevant cell models" the authors start from the consideration that the delivery of anti-inflammatory siRNA to the retinal pigment epithelium (RPE) may become a promising therapeutic option for the treatment of inflammation, if efficient delivery of siRNA to target cells is accomplished. In their study evaluates the silencing efficiency of polyplexes, lipoplexes, and lipidoid-siRNA complexes in dividing RPE cells as well as in physiologically-relevant RPE cell models. In their conclusion observe that RPE cell differentiation alters their endocytic activity and causes a decrease in the uptake of siRNA complexes. Moreover they sublimed  that melanosomal sequestration is another significant, and previously unexplored barrier to gene silencing in pigmented cells. 

In the introduction it would have been good to highlight the importance of the mesenchyme epithelium transition that occurs in physiological and pathological conditions and the molecules that regulate its variability, using as a prevailing example what occurs in angiogenesis and tumors (i.e. Scheau et al. "The Role of Matrix Metalloproteinases in the Epithelial-Mesenchymal Transition of Hepatocellular Carcinoma" 2019, Anal Cell Pathol (Amst) doi: 10.1155/2019/9423907. Ghersi "Roles of Molecules Involved in epithelial/mesenchymal Transition During Angiogenesis" 2008, Front Biosci 2335-55 doi: 10.2741/2848). As well as to consider that there are nanostructured systems that can transport biomolecules through the tissues of the eye, however, are not suitable for transporting siRNA (i.e. Di Prima et al. "Novel Inulin-Based Mucoadhesive Micelles Loaded With Corticosteroids as Potential Transcorneal Permeation Enhancers" 2017, Eur J Pharm Biopharm 385-399 doi: 10.1016/j.ejpb.2017.05.005). 

About the results, the authors should specify in the text why in Figure 3, in different conditions, there is a recovery in the secretion of IL-6 from 5 to 9 days and if this is due to a metabolic cambo or to the cell culture conditions. 

Round 2

Reviewer 3 Report

The suggested papers were aimed at the fact that other transport systems can be used and that when there is a variation at the endothelial level, a whole series of processes are activated in particular at the level of the proteolytic cascade with the generation of signal molecules.
The response of such a rapid recovery in IL-6 production, should have been monitored with a greater number of intermediate points as well as the inhibition between the second and fifth day.